# Targeted virome deep sequencing reveals frequent herpesvirus detection in intestinal biopsies of inflammatory bowel disease patients

Jennifer-Natalia Vásquez[1], Pedro Doncel[1], Juan Camacho[1], Estrella Ruiz[1], Vanessa Recio[1], David Tarragó[1,2]*

1 Centro Nacional de Microbiología, Instituto de Salud Carlos III, Madrid, Spain, 2 Institut Botànic de Barcelona, Consejo Superior de Investigaciones Científicas, Barcelona, Spain

* d.tarrago@ibb.csic.es

## Abstract

### Background

The intestinal virome is increasingly recognized for its impact on intestinal health and disease. Inflammatory bowel disease (IBD) has been linked to microbial dysbiosis, yet most studies rely on fecal samples. Here, we characterized the mucosa-associated virome directly from intestinal biopsies, providing a more localized view of viral activity at the site of pathology.

### Methods

We conducted a retrospective metagenomic study of 56 residual intestinal biopsy samples from IBD patients including ulcerative colitis (n = 37; 66.1%), IBD-Unclassified (n = 9; 16.1%), ulcerative proctitis (n = 7; 12.5%), and Crohn's disease (n = 3; 5.4%), applying high-throughput sequencing after viral nucleic acid enrichment using a probe-based capture approach. Metagenomic data were processed using the Chan Zuckerberg ID (CZ ID) platform.

### Results

Viruses were detected in 58.9% (33/56) of the biopsies, primarily members of the Herpesviridae family. EBV was the most frequently detected virus (33.9%), followed by HHV-7 (21.4%), and both CMV and HHV-6 (12.5% each), after decomposing coinfections. Other viruses such as Norovirus and human papillomavirus (HPV) were detected at lower frequencies. Coinfections were also identified. No statistically significant associations were found between viral presence and IBD (ulcerative colitis, Crohn's disease, ulcerative proctitis, and IBD-Unclassified).

**Data availability statement:** All relevant data are within the manuscript and its Supporting information files. Raw sequence data have been deposited in the European Nucleotide Archive under accession number PRJEB101152.

**Funding:** This study was supported by a grant from Instituto de Salud Carlos III. AESI2021 PCIII00011-MPY434/2021. The funder had no role in study design, data collection and analysis, decision to publish, or preparation of the manuscript.

**Competing interests:** The authors declare no conflicts of interest.

## Conclusions

Herpesviruses are rarely detected in healthy intestinal viromes and are generally considered absent, whereas their frequent presence in IBD biopsies suggests possible pathological relevance. Our findings highlight the value of metagenomic sequencing in characterizing the intestinal virome to assess the diagnostic or prognostic value of viral biomarkers in IBD.

## 1. Introduction

The human gut microbiota is a complex and dynamic ecosystem composed of trillions of microorganisms, including bacteria, archaea, fungi, and viruses. While bacterial communities have been extensively studied in the context of gastrointestinal health and disease, the viral component—collectively known as the intestinal virome has only recently begun to receive comparable attention [1–3]. The virome includes both eukaryotic viruses and bacteriophages, the latter of which play critical roles in shaping bacterial diversity and function through lytic and lysogenic cycles [4–6].

Inflammatory bowel disease (IBD), which encompasses Crohn's disease (CD), ulcerative colitis (UC), ulcerative proctitis, and unclassified IBD (IBD-U), is a chronic and relapsing inflammatory condition of the gastrointestinal tract. Although the etiology of IBD is multifactorial and not yet fully understood, it is believed to involve a complex interplay between genetic susceptibility, environmental exposures, immune dysregulation, and microbial imbalance, or dysbiosis [7,8]. While bacterial dysbiosis has been well documented in IBD, the contribution of the virome remains underexplored, particularly in clinical samples such as intestinal biopsies [2,9,10].

Recent studies suggest that shifts in the composition of the intestinal virome may be linked to intestinal inflammation [1,2,7,9,11]. In healthy individuals, the virome is dominated by lytic bacteriophages [1,4,5], while IBD patients tend to exhibit increased abundance of temperate phages [6,8,11,12]. Additionally, some studies have reported increased detection of eukaryotic viruses, including members of the Herpesviridae family, in intestinal tissue of IBD patients [13–15].

Conventional diagnostic methods for IBD lack the capacity to comprehensively characterize the virome. Metagenomic next-generation sequencing (mNGS), particularly when combined with viral nucleic acid enrichment techniques such as hybrid capture with pan-viral probes, offers a powerful approach to detect both known and novel viruses in clinical samples [9,13]. This methodology enables the identification of potential viral biomarkers that may support early diagnosis, inform prognosis, and guide personalized therapeutic strategies [16,17].

In this context, we performed a retrospective metagenomic analysis of residual intestinal biopsy samples from IBD patients, using high-throughput sequencing preceded by viral nucleic acid enrichment. Our aim was to characterize the viral diversity present in these samples and to investigate potential associations between viral detection and clinical subtypes of IBD. By leveraging an unbiased metagenomic approach, this study contributes to a deeper understanding of the intestinal virome and its possible role in the pathogenesis of IBD.

## 2. Materials and methods

### 2.1. Study design and sample collection

This was a retrospective, descriptive study conducted at the National Center for Microbiology (CNM), Instituto de Salud Carlos III, Madrid, Spain. A total of 56 residual intestinal biopsy samples were selected from the CNM, submitted for clinical diagnostic purposes between 2017 and 2023. Residual intestinal biopsies (rather than stool) were analyzed to characterize mucosa-associated viruses at the site of pathology. In addition, six negative controls consisting of nuclease-free water were included and processed in parallel throughout extraction and sequencing to monitor potential contamination. All clinical samples corresponded to patients diagnosed with inflammatory bowel disease (IBD), including ulcerative colitis (n = 37; 66.1%), IBD-Unclassified (n = 9; 16.1%), ulcerative proctitis (n = 7; 12.5%), and Crohn's disease (n = 3; 5.4%). Some patient data were accessed for research purposes during March through June 2024. Authors had not access to information that could identify individual participants during or after data collection because patient identifications were anonymized before analysis of data. Clinical data related to treatment and other patient information contained in medical records are not publicly available due to confidentiality restrictions. This study did not involve human experimentation and all methods were carried out in accordance with relevant guidelines and UE regulations. All experimental protocols including the use of residual clinical specimens submitted for virological diagnosis and written informed consent from all subjects was approved by the Ethics Committee of the "Instituto de Salud Carlos III" (CEI PI 11_2021-v3).

### 2.2. Nucleic acid extraction

Total nucleic acids were manually extracted using the QIAamp® MinElute® Virus Spin Kit (Qiagen), following the manufacturer's protocol. RNA quantification was performed with the QuantiFluor® RNA System (Promega) using a Quantus™ Fluorometer. Extracts were stored at −20°C until library preparation.

### 2.3. Library preparation and viral enrichment

Libraries were prepared using the NEBNext® Ultra II RNA Library Prep Kit (New England Biolabs) according to the manufacturer's instructions. After cDNA synthesis and adapter ligation, indexed libraries were pooled by concentration. Viral nucleic acid enrichment was performed using hybrid capture with pan-viral probes targeting 3,154 viral families (Twist Bioscience). Pools of 5–6 libraries were enriched per batch following the Twist Target Enrichment protocol, including streptavidin bead capture and post-capture amplification.

### 2.4. High-throughput sequencing

Enriched libraries were sequenced on an Illumina NovaSeq 6000 system (Illumina Inc.) at the CNM Genomics Core Facility, using a standard S4 flow cell. Raw sequence data were obtained in FASTQ format for downstream analysis. Raw sequence data were submitted to the European Nucleotide Archive (PRJEB101152).

### 2.5. Bioinformatic analysis

Metagenomic data were analyzed using the Chan Zuckerberg ID (CZID) platform (https://czid.org), an open-access, cloud-based tool for taxonomic classification of sequencing data. The analysis pipeline included preprocessing (adapter and host sequence removal). Host filtering and quality control steps include:

1. Input validation: Verification of file type (FASTQ) and number of reads. The number of reads will be capped at 75 million for single-end reads or 150 for paired-end reads.

2. ERCC removal: Removal of External RNA Controls Consortium (ERCC) sequences representing spike-in controls identified through Bowtie2.

3. QC filtering: Removal of sequencing adapters, short reads, and sequence reads with low quality (e.g., high percentage of "Ns") and low complexity regions (e.g., sequence repeats) using a customized version of fastp. More specifically, this step removes bases with quality scores below 17, reads shorter than 35 bp, reads containing low complexity regions accounting for more than 40% of the sequence, and reads containing more than 15 undetermined bases (Ns).

4. Host read removal: Removal of host-associated read sequences identified through Bowtie2 followed by HISAT2 alignments against reference host genomes.

5. Human read removal: Regardless of host, removal of human sequences identified through Bowtie2 followed by HISAT2 alignments against reference human genomes.

6. Duplicate read removal: Retention of only one representative read for read sequences that are 100% identical based on the first 70 bp using CZID-dedup.

7. Subsampling: Subsampling to 2 million paired-end reads was applied by the CZID pipeline to optimize computational resources.

Subsequently, taxonomic alignment to NCBI NT and NR databases, and quantification of metrics such as score, Z-score, reads per million (rPM), percentage identity, contig number, and E-value were analyzed. A background model was generated within the CZID pipeline using the six nuclease-free water negative controls processed in parallel. This background correction step allows the statistical comparison of each sample's read counts against the distribution observed in controls, thereby enabling computation of Z-scores and reducing false-positive taxonomic assignments due to environmental or reagent contamination. For downstream analysis of pathogens, only results with Z-score >75, percentage identity >90%, rPM > 10 and ≥1 contig were considered. mNGS pathogen detections were confirmed by multiplex real-time PCR as previously described by Recio et al. [18].

## 2.6. Statistical analysis

To assess whether specific viral detections were associated with intestinal disease type, we performed a Fisher–Freeman–Halton permutation test for each virus, using 2 × 4 contingency tables representing presence/absence × disease category (*ulcerative colitis*, *Crohn's disease*, *ulcerative proctitis*, and *IBD-U*).

Samples containing multiple viral species were expanded so that each virus contributed independently, ensuring that coinfections were correctly counted.

Permutation-based p-values were computed using 20,000 Monte Carlo iterations, obtained by randomly permuting disease labels while preserving virus-specific positivity rates. A global (omnibus) statistic was calculated as the sum of the $\chi^2$ values across all viruses, and its significance was assessed through the same permutation framework.

False discovery rate (FDR) correction was applied to the individual *p*-values using the Benjamini–Hochberg method.

## 3. Results

### 3.1. Patient demographics and disease distribution

The cohort included patients clinically diagnosed with ulcerative colitis (n = 37, 66.1%), Crohn's disease (n = 3, 5.4%), ulcerative proctitis (n = 7, 12.5%), and IBD-U (inflammatory bowel disease, unclassified) (n = 9, 16.1%). The median patient age was 46 years (range, 12–91), and 57.6% (n = 19) were male. No statistical power calculation was performed to predetermine sample size, as the cohort represented the complete set of available samples during the study period. This approach ensures that the results reflect the full diagnostic spectrum accessible for analysis, although the limited sample size is acknowledged as a study limitation.

## 3.2. Viral detection in intestinal biopsies

A total of 56 intestinal biopsy samples were analyzed, of which viral sequences were identified in 33 samples (58.9%), while the remaining 23 showed no detectable viral signal. When coinfections were decomposed into individual viral counts, most of identified viruses belonged to the Herpesviridae family including EBV in 19/56 (33.9%), HHV-7 in 12/56 (21.4%), and CMV and HHV-6 each in 7/56 (12.5%). Fig 1 shows distribution of herpesvirus detections by intestinal disease type. Average reads per sample were 32.854.127 and average reads passing filters per sample were 801.876, including negative controls. The resulting taxonomic assignments correspond to viral taxa that passed the internal background modeling step, which statistically corrects each sample's signal against six nuclease-free water negative controls processed in parallel. S1 Table in S1 File summarizes all taxa that remained after background modeling, including their respective scores, Z-scores, reads per million (rPM), and percentage identity values. Coinfections were also observed in several samples (Fig 2). Other viruses such as Norovirus and human papillomavirus (HPV) were detected at lower frequencies. However, only pathogens with Z-score >75, percentage identity >90%, rPM > 10 and ≥1 contig were considered. These thresholds follow the recommended confidence filters established for the Chan Zuckerberg ID (CZID) metagenomic pipeline, which combine multiple metrics to minimize false-positive taxonomic assignments. The Z-score compares observed read counts to background levels from negative controls, where values > 75 indicate a highly significant enrichment unlikely due to random noise. A percentage identity above 90% ensures reliable sequence homology with reference genomes, while rPM > 10 reduces the influence of low-abundance taxa near the detection limit. The requirement of at least one assembled contig further increases specificity by confirming consistent genomic context. When applying these filters to our dataset mainly pathogens of the *Herpesviridae* family were retained, indicating that the thresholds effectively

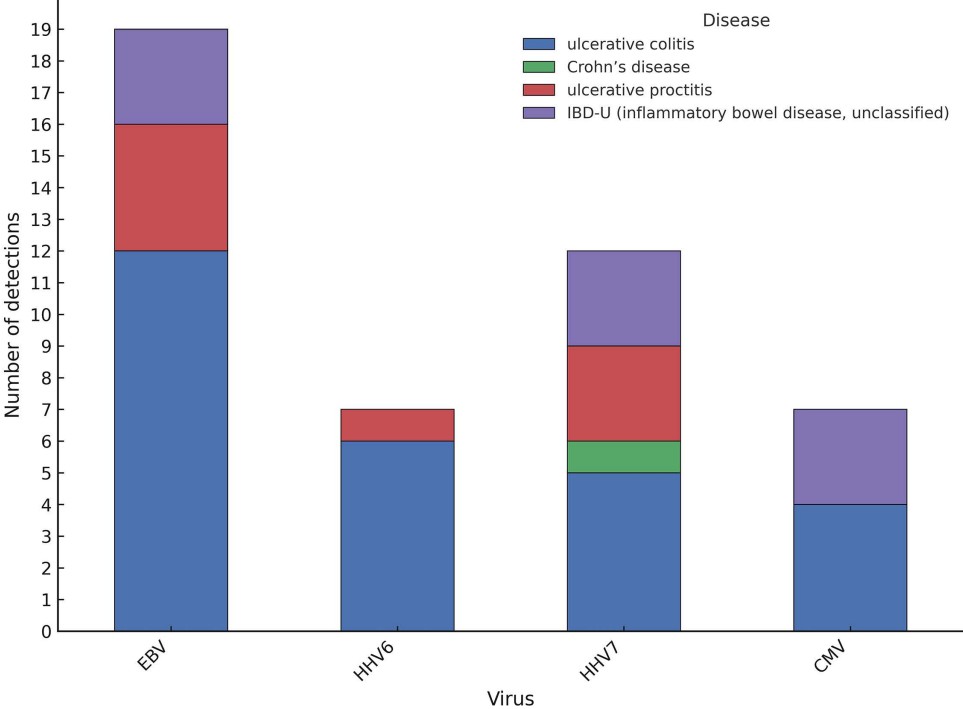

**Fig 1. Distribution of herpesvirus detections by intestinal disease type.** Stacked bar plot showing the number of viral detections per virus across intestinal disease categories: *ulcerative colitis*, *Crohn's disease*, *ulcerative proctitis*, and *IBD-U (inflammatory bowel disease, unclassified)*. Each bar represents the total number of samples positive for a given virus, with coinfections counted independently for each viral species. Y-axis indicates the absolute number of detections per virus.

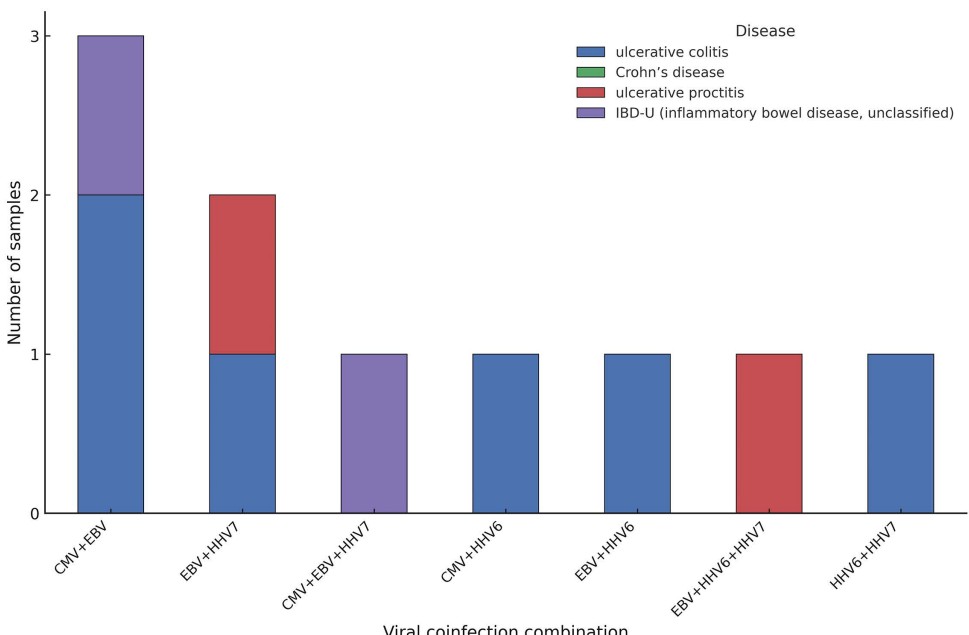

**Fig 2. Viral coinfection combinations across intestinal disease types.** Stacked bar plot showing the frequency of distinct viral coinfection combinations detected in intestinal biopsies from patients with *ulcerative colitis*, *Crohn's disease*, *ulcerative proctitis*, and *IBD-U (inflammatory bowel disease unclassified)*. Each bar represents one unique combination of two or more viral species, treated as identical regardless of detection order (e.g., *CMV + EBV = EBV + CMV*). All values on the Y-axis correspond to the absolute number of samples identified for each coinfection combination.

removed spurious low-signal taxa and captured biologically meaningful viral reads. Detailed per-sample metrics after applying quality thresholds are provided in S2 Table in S1 File. Similar criteria for post-processing and hit validation have been used in prior CZID-based metagenomic studies [19,20].

No virus showed a statistically significant difference in distribution across disease categories after FDR correction (*all q > 0.05*). Detailed counts per virus and disease category, together with individual Fisher–Freeman–Halton permutation p and FDR-adjusted q values, are provided in S3 Table in S1 File.

### 3.3. Association between viral detection and IBD subtype

After accounting for coinfections, no virus showed a statistically significant difference in detection frequency across disease categories (*all q[FDR] > 0.05*).

The global (omnibus) comparison yielded a summed $\chi^2$ statistic of 22.51, with a permutation $p = 0.2473$, indicating no significant overall association between the intestinal viral profile and disease type in this cohort.

### 3.4. Bacterial and phage detection

In addition to viruses and despite viral enrichment, bacterial sequences were identified in 25 of the 56 samples (44.6%). The most common bacterial taxa included members of the Enterobacteriaceae, Streptococcaceae, and Bacteroidaceae families, particularly Klebsiella spp. and Escherichia coli. Their presence supports previously described associations between dysbiosis and intestinal inflammation. Several sequences aligned with bacteriophages, mainly from the Caudoviricetes class. Though not the primary target of the study, these findings support existing evidence of increased temperate phages in IBD and suggest ecological interactions that may influence mucosal homeostasis.

## 4. Discussion

In this study, we performed a metagenomic characterization of the intestinal virome in biopsy samples from patients with IBD using high-throughput sequencing preceded by viral nucleic acid enrichment. We prioritized intestinal biopsies rather than stool because our primary question concerned mucosa-associated viral activity and its relationship with local pathology. Tissue sampling (i) captures site-specific, epithelial/lamina propria–associated viruses that may be under-represented or undetectable in bulk stool; (ii) enables direct clinico-pathological correlation (histology, inflammation grade, ulceration) at the exact site of detection; (iii) reduces confounding from luminal transit, diet, and environmental contamination that complicate stool virome profiles; and (iv) increases specificity for latent/reactivating herpesviruses, which may persist in tissue niches (e.g., B-cell infiltrates) without robust luminal shedding. While stool is valuable for population-level surveillance and phage ecology, it can dilute signals of tissue-resident DNA viruses and blur spatial resolution [21]. Using residual diagnostic biopsies allowed standardized processing, parallel negative controls, and alignment with histopathology to address whether herpesvirus detection reflects local mucosal processes rather than incidental passage.

Our results demonstrate a high prevalence of viruses—particularly herpesviruses—within intestinal mucosa, despite the absence of statistically significant associations with IBD subtypes. Epstein–Barr virus (EBV) was the most frequently detected pathogen, present in more than 57% of positive intestinal biopsy samples. This observation is consistent with previous studies reporting EBV persistence within intestinal mucosa, particularly among immunocompromised individuals and patients with inflammatory bowel disease (IBD) [14,15]. EBV establishes latency in B lymphocytes and can be reactivated under inflammatory or immunosuppressive conditions, leading to local expansion of infected cells and increased cytokine production. Several reports have proposed that latent EBV infection may exacerbate intestinal inflammation through modulation of mucosal immune responses and epithelial integrity, especially in ulcerative colitis, where EBV-positive lymphocytes have been detected within inflamed lamina propria and ulcerated areas [22–24]. Whether EBV acts as a primary etiological agent or as a cofactor in intestinal pathology remains under debate. Current evidence indicates that EBV is rarely a primary cause of intestinal disease; rather, it tends to exploit inflamed or immunocompromised tissue as a niche for reactivation. In patients with ulcerative colitis and other forms of IBD, EBV reactivation has been associated with increased mucosal inflammation, higher disease activity, and resistance to immunosuppressive therapy, suggesting a role as an inflammatory amplifier rather than an initiating pathogen. Nevertheless, in severely immunosuppressed individuals—such as transplant recipients or patients with lymphoproliferative disorders—EBV-driven mucosal lesions or lymphoid infiltrates may represent a more direct pathogenic mechanism.

Human herpesviruses 6 and 7 (HHV-6, HHV-7) and cytomegalovirus (CMV) were also commonly identified. CMV, in particular, has been implicated in disease flares and steroid-refractory colitis [24], but again, our findings did not reveal a significant association with IBD subtype. HHV-6 and HHV-7 are frequently found in various tissues and may represent bystander infections; nonetheless, their potential role as cofactors in chronic mucosal inflammation warrants further investigation [24]. The presence of viral DNA from herpesviruses in intestinal biopsies has been reported primarily in association with mucosal inflammation, epithelial disruption, or local immune dysregulation, particularly in immunocompromised or IBU patients. Therefore, their detection in intestinal tissue is more consistent with pathological reactivation or opportunistic replication than with a commensal virome component.

This interpretation aligns with previous studies showing that herpesvirus sequences are rarely or not detected in the fecal or mucosal virome of healthy controls, but become evident in disease-associated contexts [11,25,26].

Our finding of multiple coinfections, including EBV + CMV and EBV + HHV-7, is notable and suggests that viral interactions within the gut mucosa may be more complex than previously appreciated [17]. Such coinfections are increasingly recognized in immunocompromised hosts and patients with chronic inflammatory conditions, where overlapping viral reactivations can occur under systemic or local immune suppression. Experimental and clinical data indicate that herpesviruses may influence one another's replication dynamics through shared latency and reactivation pathways, cytokine induction, or competition for cellular niches. In the intestinal mucosa, concurrent reactivation of EBV and CMV has been

associated with more severe histological inflammation, increased epithelial apoptosis, and reduced responsiveness to immunomodulatory therapy. Whether these co-detections contribute synergistically to mucosal injury, or instead reflect a broader state of immune dysregulation and loss of viral control in IBD, remains to be determined. Future longitudinal studies integrating host immune profiling and viral transcriptional activity will be necessary to disentangle causality from consequence in these complex viral interactions.

Beyond viral pathogens, bacterial taxa were also detected in 25 of the 56 samples, despite the probe-based enrichment being designed to target viruses. Identified bacteria included commensal species and potentially pathogenic taxa, especially members of the Enterobacteriaceae family such as Klebsiella spp. These findings are consistent with prior reports linking Enterobacteriaceae overgrowth with mucosal inflammation in IBD [3,12].

Additionally, although our analysis was not optimized for bacteriophage detection, several sequences aligned to phages, particularly from the Caudoviricetes class. Prior studies have shown that an increased abundance of temperate phages and reduced phage diversity are hallmarks of IBD-associated dysbiosis [4,6,8,11]. Phages may modulate bacterial composition, contribute to immune activation, and indirectly influence intestinal inflammation. The co-detection of phages and pathogenic bacteria in the same samples may indicate ecological interactions relevant to disease activity [4,5,23].

Importantly, our methodology focused on virome characterization through a probe-based enrichment protocol targeting a wide array of viral families. This approach, combined with metagenomic sequencing, enabled the detection of low-abundance and highly divergent viral genomes [13,17]. Nevertheless, it introduces potential biases, particularly against viruses present at very low titers or with insufficient probe homology [13]. The absence of a healthy control group limits our ability to determine whether the viruses detected are enriched in IBD compared to healthy individuals. Therefore, without a baseline, caution is warranted when interpreting their role in disease pathogenesis.

Unlike enteric bacteriophages or endogenous anelloviruses, members of the *Herpesviridae* family are typically absent from the healthy intestinal virome and are instead detected transiently during systemic or mucosal reactivation events [9,14,15].

Our study highlights the utility of metagenomic sequencing in detecting and characterizing the intestinal virome directly from clinical samples. While current findings do not support a specific viral signature differentiating IBD subtypes, the frequent presence of herpesviruses—particularly EBV—supports their inclusion in future research exploring viral biomarkers or potential triggers of disease activity [14,22,24].

## 5. Conclusions

This study demonstrates the frequent detection of human herpesviruses, particularly Epstein-Barr virus, human herpesvirus 7, and cytomegalovirus, in intestinal biopsies from patients with inflammatory bowel disease. Although no statistically significant associations were found between viral presence and IBD subtype, the consistent detection of these viruses—absent from the healthy intestinal virome [9,14,15]—raises questions about their potential involvement in disease progression or immune modulation. The incidental identification of bacterial taxa and bacteriophages further supports the existence of a complex microbial network influencing gut inflammation [4,12]. Our findings highlight the usefulness of metagenomic sequencing with viral enrichment in exploring the intestinal virome and call for further studies including healthy controls to validate viral biomarkers in IBD [13,17].

## Supporting information

**S1 File. Supplementary tables related to virome analysis.** This file contains the following supplementary tables referenced in the manuscript: **S1 Table.** Virome characteristics (CZID metrics) of 56 intestinal biopsy samples from patients with inflammatory bowel disease (IBD). **S2 Table.** CZID metrics per sample after quality thresholds. **S3 Table.** Viral detection by disease type and Fisher–Freeman–Halton permutation p-values.
(PDF)

## Acknowledgments

We are very grateful to Sergi Arteaga for preparing the manuscript and to María Dolores Pérez-Vázquez from the Centro de Investigación Biomédica en Red de Enfermedades Infecciosas (CIBERINFEC) for her kind collaboration at different stages of the study.

## Author contributions

**Conceptualization:** David Tarragó.

**Data curation:** Jennifer-Natalia Vasquez, Pedro Doncel.

**Formal analysis:** David Tarragó.

**Funding acquisition:** David Tarragó.

**Investigation:** David Tarragó.

**Methodology:** Jennifer-Natalia Vasquez, Juan Camacho, Vanessa Recio, Estrella Ruiz.

**Validation:** David Tarragó.

**Writing – original draft:** Jennifer-Natalia Vasquez.

**Writing – review & editing:** David Tarragó.

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
