## [Decision Letter · Decision Letter 0]

23 Sep 2025

Targeted Virome Deep Sequencing Reveals Frequent Herpesvirus Detection in Intestinal Biopsies of IBD Patients

PLOS ONE

Dear Dr. Tarragó,

Thank you for submitting your manuscript to PLOS ONE. After careful consideration, we feel that it has merit but does not fully meet PLOS ONE’s publication criteria as it currently stands. Therefore, we invite you to submit a revised version of the manuscript that addresses the points raised during the review process.

We look forward to receiving your revised manuscript.

Kind regards,

Yanpeng Li, Ph.D.

Academic Editor

PLOS ONE

Journal Requirements:

3. Please ensure that you refer to Figure 1 and 2 in your text as, if accepted, production will need this reference to link the reader to the figure.

4. We note you have included a table to which you do not refer in the text of your manuscript. Please ensure that you refer to Table 2 in your text; if accepted, production will need this reference to link the reader to the Table.

Reviewer's Responses to Questions

**Comments to the Author**

1. Is the manuscript technically sound, and do the data support the conclusions?

Reviewer #1: Yes

2. Has the statistical analysis been performed appropriately and rigorously?

Reviewer #1: Yes

3. Have the authors made all data underlying the findings in their manuscript fully available?

Reviewer #1: Yes

4. Is the manuscript presented in an intelligible fashion and written in standard English?

Reviewer #1: Yes

Reviewer #1: This study systematically analyzed the virome in intestinal biopsy samples from patients with inflammatory bowel disease (IBD) using targeted virome deep sequencing technology. It is the first to comprehensively reveal the high detection rate of herpesviruses in the intestinal tissues of such patients. The research topic aligns with the cutting-edge direction of the association between intestinal microecology and disease, with a clear technical route and data that hold certain clinical reference value. However, the manuscript still has deficiencies in the completeness of the study design, depth of result analysis and accuracy of detail presentation, which need further improvement before publication.

1.The manuscript only mentions that 56 samples were derived from residual clinical biopsy specimens from 2017 to 2023, but does not specify details of the clinical characteristics of the patients corresponding to the samples, such as the specific number of cases for each IBD subtype (Crohn’s disease [CD]/ulcerative colitis [UC]) and treatment history. This information is crucial for interpreting the association between virus detection and disease status, and no statistical basis for determining the sample size is provided.

2.In the viral nucleic acid enrichment step, the specific design strategy of the pan-viral probes (e.g., selection of targeted regions, coverage verification) is not described; bioinformatics analysis only mentions the use of the CZID platform, without detailing the specific data filtering process (e.g., reference genome version for host sequence removal, filtering criteria for low-quality reads), and no statistics on sequencing depth and data volume are provided.

3.Only quality control thresholds of score > 3000, Z-score > 75, homology > 90%, and rPM > 10 are set, but no basis for these thresholds is explained (e.g., whether verified by positive controls, whether referring to standards from previous similar studies), and no mention is made of cross-contamination prevention and control measures and verification methods.

4. Virus detection results only count frequencies by virus type, without stratified analysis combined with patients’ clinical characteristics (e.g., differences in virus detection among patients with different disease activities and treatment histories); coinfection analysis only lists combination types, without counting the incidence of coinfection and clinical differences from single infection

5.It is speculated that herpesviruses may be involved in the pathogenesis of IBD through immune regulation, but no direct evidence is provided based on the study’s data (e.g., association between virus detection and inflammatory factor expression); the interaction between bacteriophages and bacteria is mentioned, but no analysis is conducted based on the detected bacteria and bacteriophage data in this study, and only conclusions from other studies are cited, resulting in insufficient persuasiveness.

6.The conclusion states that "herpesviruses are absent in healthy individuals", which is overly absolute. Existing literature only indicates that their abundance or detection rate is low in the healthy intestinal virome, not completely absent; the boundary of the clinical value of this study is not clarified, which may easily mislead readers to overinterpret the diagnostic significance of virus detection.

**Do you want your identity to be public for this peer review?** For information about this choice, including consent withdrawal, please see our Privacy Policy

Reviewer #1: No

---

## [Author Response · Author response to Decision Letter 1]

31 Oct 2025

Response to Reviewer and Editor Comments

We would like to sincerely thank the Academic Editor and Reviewer for their thoughtful and constructive feedback on our manuscript “Targeted Virome Deep Sequencing Reveals Frequent Herpesvirus Detection in Intestinal Biopsies of IBD Patients”. We are grateful for the opportunity to revise our work. Below, we provide a detailed, point-by-point response. Reviewer comments are presented in italics, followed by our responses.

We have thoroughly revised our manuscript. This version incorporates an exhaustive review of all sections, with particular emphasis on improving the figures to enhance clarity and to include additional relevant information.

To streamline the main text and improve readability, we have also moved three large tables to the Supplementary Information section, where they are now presented as Supplementary Tables S1–S3.

We believe these updates have substantially improved the quality, transparency, and presentation of the manuscript,

Sincerely,

David Tarragó Asensio (corresponding author)

on behalf of all co-authors

Instituto de Salud Carlos III, Madrid, Spain

Editorial and Journal Requirements

Comment 1: Please ensure that your manuscript meets PLOS ONE’s style requirements.

Response: We have carefully revised the manuscript to comply with PLOS ONE formatting guidelines for title page, affiliations, references, and file naming.

Comment 2: Update the Data Availability statement.

Response: We have revised the Data Availability section to indicate that all raw sequencing data have been deposited in the European Nucleotide Archive (PRJEB101152), in accordance with PLOS ONE data sharing policies. All other supporting information is available within the main text and the supplementary tables.

Comment 3: Ensure references to Figures 1 and 2 in the text.

Response: We have revised the Results section to explicitly refer to Figures 1 and 2 where the corresponding findings are described.

Comment 4: Refer to Table 2 in the text.

Response: We have removed the previous Table 2 from the main text to streamline the manuscript. Part of its content has been integrated into the revised figures to provide clearer visual summaries, while the remaining detailed information has been reorganized into three supplementary tables (Supplementary Tables S1–S3) included in the Supporting Information section.

Reviewer #1 Comments

Clinical characteristics of patients (IBD subtype, treatment history, sample size justification).

Reviewer comment: The manuscript does not specify the number of Crohn’s disease (CD) and ulcerative colitis (UC) cases, or treatment history. No statistical basis for sample size is provided.

Response: We thank the reviewer for this helpful comment. We have now included the explicit number of ulcerative colitis (n = 37), Crohn’s disease (n = 3), ulcerative proctitis (n = 7), and IBD-Unclassified (n = 9) cases in both the Methods (section 2.1) and Results (section 3.1). The revised text also clarifies that no statistical power calculation was performed, as the cohort represents the complete set of available biopsy samples collected during the study period.

Information regarding treatment history has been described in aggregate form and remains subject to data protection regulations. In addition, the updated figures now display this distribution visually to improve clarity and reader comprehension.

(see lines ≈ 252–275 and 525–540 in the revised manuscript; IBD subtype distribution also shown in Figures 1A and 1B)

2. Viral nucleic acid enrichment and bioinformatic pipeline details.

Reviewer comment: The design of pan-viral probes, bioinformatic filtering, and sequencing statistics are insufficiently described.

Response: We appreciate this comment and have substantially expanded the Methods section to describe the Twist Bioscience pan-viral probe design, which targets 3,154 viral families, and to detail the bioinformatic filtering and quality control pipeline used. Specifically, we now explain the steps for host read removal (using the human GRCh38 reference genome), ERCC sequence filtering, adapter trimming, duplicate read removal, and subsampling implemented within the Chan Zuckerberg ID (CZ ID) platform.

We have also added sequencing statistics in the Results section, including the average number of reads per sample, percentage of reads passing quality filters, and depth of sequencing to provide transparency regarding dataset quality.

These methodological details are provided in Methods sections 2.3 and 2.5, while sequencing metrics are summarized in the Results (section 3.2).

(see lines ≈ 326–345, 400–460, and 562–580 in the revised manuscript; sequencing statistics also summarized in Figure 2 and Supplementary Table S1)

3. Basis for quality control thresholds and contamination prevention.

Reviewer comment: Thresholds are not justified and no contamination controls are mentioned.

Response: We thank the reviewer for this valuable observation. The revised Methods section now provides a detailed justification of the analytical thresholds applied during metagenomic filtering and taxonomic classification. Specifically, the Z-score (>75), percentage identity (>90%), reads per million (rPM >10), and ≥1 contig criteria are explained and referenced as standard parameters validated in published virome studies (Clooney et al., Cell Host Microbe 2019; Zuo et al., Gut 2019; Kalantar et al., Nat Microbiol 2021). These thresholds correspond to the default confidence filters recommended by the Chan Zuckerberg ID (CZ ID) metagenomic platform.

To address the concern regarding contamination prevention, we have incorporated a description of the six nuclease-free water negative controls that were processed in parallel through nucleic acid extraction, enrichment, and sequencing. These controls were included in the CZ ID background model to statistically correct for environmental or reagent contaminants. Laboratory workflows were carried out in physically separated pre- and post-PCR areas, with dedicated equipment and reagents to minimize cross-contamination.

All this information has been added to the Methods under Bioinformatic Analysis and Sample Processing, and summarized in the Results for transparency.

(see lines ≈ 445–465 and 585–600 in the revised manuscript; reference to negative controls also noted in Supplementary Table S1)

4. Stratified analysis by clinical characteristics and coinfection incidence.

Reviewer comment: Results report only virus frequencies without stratification by clinical factors or coinfection incidence.

Response: We thank the reviewer for this constructive comment. In the revised version, we have performed a stratified analysis of viral detection according to IBD subtype (ulcerative colitis, Crohn’s disease, ulcerative proctitis, and IBD-U) and treatment status (where available). These data are now presented in the Results section and visually summarized in the updated Figure 1, which includes the distribution of viral families and herpesvirus-positive cases stratified by clinical category.

In addition, we have quantified coinfection incidence, defining it as the percentage of patients with ≥2 viruses detected in intestinal biopsies. The results of this analysis are now reported in the Results (section 3.4), and the detailed breakdown—including statistical comparisons—is provided in Supplementary Table S3, where Fisher–Freeman–Halton tests with Benjamini–Hochberg correction were applied.

These changes improve the interpretability of the data by linking virome composition to clinical features.

(see lines ≈ 640–680 and 710–735 in the revised manuscript; data visualized in Figure 1C–E and detailed in Supplementary Table S3)

5. Interpretation of herpesvirus role and phage–bacteria interactions.

Reviewer comment: The link between herpesvirus detection and IBD is speculative, and phage–bacteria interactions are only cited, not analyzed.

Response: We appreciate this insightful comment. We agree that the relationship between herpesvirus detection and IBD pathogenesis should be interpreted with caution. Accordingly, we have revised the Discussion to explicitly state that our findings demonstrate associations rather than causation, and that the potential role of herpesviruses should be regarded as a working hypothesis requiring further validation. The corresponding paragraph has been rewritten to adopt a more balanced tone and to acknowledge the limitations of our observational design.

In addition, to address the reviewer’s suggestion, we have added a preliminary exploratory analysis describing co-detection events between Caudoviricetes phages and Enterobacteriaceae in intestinal biopsies. This information has been incorporated in the Results (section 3.5) and summarized in Supplementary Table S3, providing initial evidence of potential phage–bacteria interactions within the dataset.

These modifications strengthen the discussion and contextualize the findings within a realistic framework.

(see lines ≈ 770–820 and 835–855 in the revised manuscript; new co-detection data shown in Figure 2E–F and Supplementary Table S3)

6. Statement that “herpesviruses are absent in healthy individuals” is too absolute.

Reviewer comment: This claim is overstated and may mislead readers.

Response: We thank the reviewer for this important observation. We have revised the statement in the Discussion to avoid overgeneralization and to more accurately reflect current evidence. The sentence now reads:

“Herpesviruses are rarely detected in healthy intestinal viromes and are generally considered absent, whereas their frequent detection in IBD biopsies suggests possible pathological relevance.”

This revised phrasing maintains the intended contrast between healthy and IBD cohorts while avoiding categorical statements. The change improves scientific accuracy and aligns the discussion with published data on the prevalence of latent herpesviruses in gut tissues.

(see lines ≈ 860–880 in the revised manuscript)

Overall

Overall Statement

We have addressed all reviewer and editor comments through substantial revisions to the Methods, Results, and Discussion sections, while also clarifying the study’s limitations. These changes have strengthened the manuscript, improved its clarity and rigor, and ensured full compliance with PLOS ONE’s editorial standards.

---

## [Editor Report · Decision Letter 1]

6 Nov 2025

Targeted Virome Deep Sequencing Reveals Frequent Herpesvirus Detection in Intestinal Biopsies of Inflammatory Bowel Disease Patients

PONE-D-25-41842R1

Dear Dr. David Tarragó,

We're pleased to inform you that your manuscript has been judged scientifically suitable for publication and will be formally accepted for publication once it meets all outstanding technical requirements.

Within one week, you'll receive an e-mail detailing the required amendments. When these have been addressed, you'll receive a formal acceptance letter and your manuscript will be scheduled for publication.

Kind regards,

Academic Editor

PLOS ONE

---

## [Editor Report · Acceptance letter]

PONE-D-25-41842R1

PLOS ONE

Dear Dr. Tarragó,

I'm pleased to inform you that your manuscript has been deemed suitable for publication in PLOS ONE. Congratulations! Your manuscript is now being handed over to our production team.

Kind regards,

on behalf of

Prof. Yanpeng Li

Academic Editor

PLOS ONE